# Subducted organic matter buffered by marine carbonate rules the carbon isotopic signature of arc emissions

S. Tumiati [1] [✉], S. Recchia [2], L. Remusat[3], C. Tiraboschi[1,4], D. A. Sverjensky[5], C. E. Manning[6], A. Vitale Brovarone [7], A. Boutier[8], D. Spanu[2] & S. Poli [1]

Ocean sediments consist mainly of calcium carbonate and organic matter (phytoplankton debris). Once subducted, some carbon is removed from the slab and returns to the atmosphere as $CO_2$ in arc magmas. Its isotopic signature is thought to reflect the bulk fraction of inorganic (carbonate) and organic (graphitic) carbon in the sedimentary source. Here we challenge this assumption by experimentally investigating model sediments composed of $^{13}C$-$CaCO_3$ + $^{12}C$-graphite interacting with water at pressure, temperature and redox conditions of an average slab–mantle interface beneath arcs. We show that oxidative dissolution of graphite is the main process controlling the production of $CO_2$, and its isotopic composition reflects the $CO_2$/$CaCO_3$ rather than the bulk graphite/$CaCO_3$ (i.e., organic/inorganic carbon) fraction. We provide a mathematical model to relate the arc $CO_2$ isotopic signature with the fluid–rock ratios and the redox state in force in its subarc source.

[1] Dipartimento di Scienze della Terra, Università degli Studi di Milano, via Mangiagalli 34, I-20133 Milano, Italy. [2] Dipartimento di Scienza e Alta Tecnologia, Università degli Studi dell'Insubria, via Valleggio 11, I-22100 Como, Italy. [3] Institut de Minéralogie, de Physique des Matériaux, et de Cosmochimie (IMPMC), Sorbonne Universités – UPMC, UMR CNRS, 7590, Muséum National d'Histoire Naturelle, IRD UMR 206, F-75005 Paris, France. [4] Institut für Mineralogie, Universität Münster, Correnstrasse 24, 48149 Münster, Germany. [5] Department of Earth & Planetary Sciences, Johns Hopkins University, Baltimore, MD 21218, USA. [6] Department of Earth, Planetary and Space Sciences, University of California, Los Angeles, CA 90095-1567, USA. [7] Dipartimento di Scienze Biologiche, Geologiche e Ambientali (BiGeA), Alma Mater Studiorum Università di Bologna, 40126 Bologna, Italy. [8] Dipartimento di Scienze della Terra, Università degli Studi di Torino, via Valperga Caluso 35, 10125 Torino, Italy. [✉]email: simone.tumiati@unimi.it

Modern open-ocean sediments are dominated by phytoplankton. Calcium carbonate, chiefly in sediments deposited above the calcite compensation depth, is essentially calcite forming the shells of organisms such as coccolithophores. It displays carbon isotopic ratios comparable to bicarbonate ions dissolved in seawater, characterized by $\delta^{13}C \approx 0‰$ expressed as $^{13}C/^{12}C$ per mil difference normalized to the international Vienna-Pee Dee Belemnite (VPDB) standard. Conversely organic matter, which is essentially phytoplankton debris in the open seafloor, is depleted at $^{13}C$ due to fractionation effects induced by photosynthesis, such that $\delta^{13}C \approx -20‰$ VPDB[1]. Stable carbon isotopes and mass-balance calculations relying on simple mixing models have been used extensively to determine the relative contribution of organic matter and marine carbonates in sedimentary rocks[2], in their metamorphic equivalents[3] and in volcanic arc emissions[4]. However, the applicability of simple mixing models hinges on assumptions that may be overly simplistic, including that the sedimentary "end-member" compositions are spatially and temporally invariant, and that isotopic equilibrium is attained during metamorphism. The latter is particularly problematic because sedimentary organic carbon and coexisting carbonate may exhibit substantial isotopic exchange with increasing metamorphic grade, in particular at temperatures >650 °C during prograde graphitization[3,5–7]. In turn, fully crystalline graphite displays a sluggish rate of isotopic diffusion and may be unaffected by isotopic reset even in cases of intense metamorphism and fluid/rock interactions[8].

Carbonates and graphitic carbon derived from organic matter, formerly assumed to be refractory to dissolution and devolatilization during subduction[9,10], are now thought to show a non-negligible solubility in subduction fluids at least at certain $P–T–fO_2–pH$ conditions[11–18]. The interaction of subducted sediments with deep fluids[14,19] produces dissolved carbon that is transferred from the slab to the overlying mantle wedge, prompting carbonation/metasomatism and/or partial melting[20–23], and eventually returning to the surface via $CO_2$ emitted by arc volcanoes[4]. The global arc average $\delta^{13}C$ is $-2.8$ to $-3.3$ ‰[4]. This is heavier than the major carbon isotopic composition signature of the upper mantle ($\delta^{13}C = -6.0 \pm 2.5‰$)[4], but significantly lighter than sedimentary carbonates ($\delta^{13}C \approx 0‰$[24]). As it is generally assumed that the carbon isotope signature of arc emissions reflects that of their source, relatively high $\delta^{13}C$ values would point out assimilation of shallow "crustal" limestones, while low $\delta^{13}C$ are usually attributed to subducted organic carbon[4]. However, processes of dissolution and of isotopic exchange involving organic and inorganic carbon beneath arcs are still not fully understood.

In this work, we investigate the carbon isotopic exchange occurring in a model system representative of open-ocean sediments containing calcium carbonate + organic matter subducted at subarc depths and interacting with aqueous fluids rising from the underlying dehydrating oceanic lithosphere[25]. We provide the quantitative chemical analysis of the volatile species and the measured carbon isotopic composition of $CO_2$ produced by dissolution in water of graphite and of aragonite at $P = 3$ GPa, $T = 700$ °C and at redox-controlled conditions buffered to $fH_2 =$ FMQ (equivalent to $fO_2$ expressed as $\Delta$FMQ $= +0.61$ log units). These conditions are selected on the basis of the predicted peak of $CO_2$ produced by oxidative dissolution of graphite in subduction zones (Supplementary Fig. 1). Starting materials of synthetic labelled $CaCO_3$ (99.4% $^{13}C$) and synthetic graphite (98.9% $^{12}C$) are used as analogues for natural "heavy" carbonate and "light" organic matter and to generate a maximum isotopic difference in experiments (close to pure $^{13}C$ and $^{12}C$ end-members). Control experiments are performed with oxalic acid di-hydrate (98.7% $^{12}C$) as the source of $CO_2$ instead of graphite. The dissolution process and the effect of different $CO_2$/aragonite ratios and of different run durations from 0.24 to 240 h are evaluated and compared with thermodynamic calculations that include consideration of aqueous solute speciation[26–28] or consider only gas speciation in a conventional graphite-saturated COH fluid model[29,30]. Finally, a comparison with conventional isotopic models is provided. Experimental data allow the development of mathematical models extending the applicability of the results to a wide range of redox conditions and of fluid/rock ratios in order to predict the $\delta^{13}C$ of $CO_2$ released from the sedimentary slab and to envisage conditions required to meet the global average arc signature.

## Results and discussion

**$CO_2$ evolved by the aqueous dissolution of aragonite-only, graphite-only and mixed aragonite + graphite.** In all experiments we observed that aragonite crystals in run products displayed textural evidence of dissolution-reprecipitation, including step edges, and fine-grained recrystallized rims around larger relict cores (Fig. 1). Conversely, we never observed texturally precipitation of newly formed graphite or graphite recrystallization, nor evidence for graphite isotopic variation compared to its starting composition.

In aragonite-only runs, despite the evidence of dissolution–precipitation microtextures, the absolute amount of $CO_2$ evolved by the aqueous dissolution of aragonite-only is very low, close to the analytical detection limit, with $XCO_2$ $[=CO_{2(aq)}/(H_2O + CO_{2(aq)})_{molar}] = 0.001$ corresponding to 0.041 mol $CO_2$/kg $H_2O$ (Fig. 2a; Supplementary Tables 1 and 2). We compared this result with thermodynamic modelling performed at the investigated $P–T–fO_2$ conditions using the Deep Earth Water (DEW)

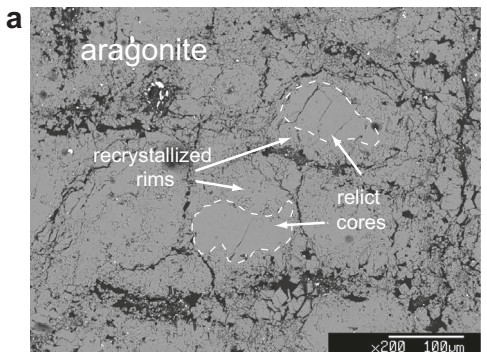
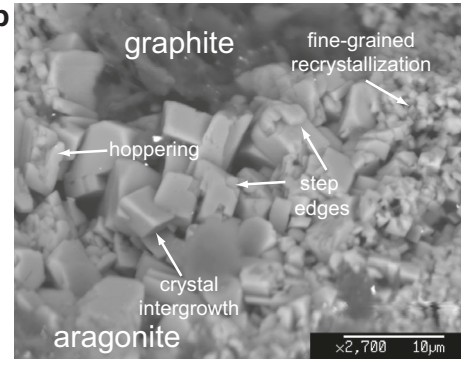

**Fig. 1 Electron microscope images showing microtextures of run products. a** Recrystallized aragonite, showing rims of sub-micrometric crystals surrounding relict cores 50–100 μm in size. **b** Assemblage graphite + aragonite; aragonite crystals show dissolution/reprecipitation microtextures such as crystal size reduction, hoppering, step edges and euhedral crystal intergrowth.

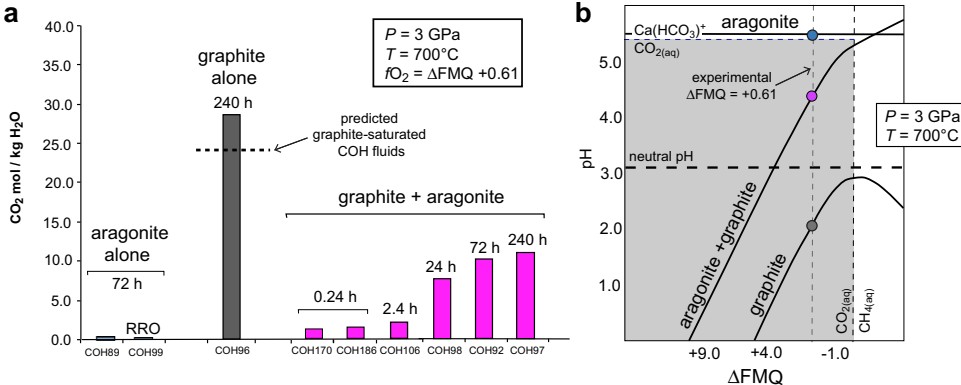

**Fig. 2 Experimental CO₂ contents and thermodynamic modelling of fluids. a** $CO_2$ concentration (molality) in experimental aqueous fluids interacting with (i) aragonite-only (blue), (ii) graphite-only (grey) and (iii) graphite + aragonite (magenta). Typical analytical uncertainty is 1 mol%. Dashed line: $CO_2$ content predicted by thermodynamic modelling of graphite-saturated COH fluids. Run duration (h) is shown at the top of each bar. RRO: run buffered by Re–ReO₂ ($\Delta FMQ \approx +2$) instead of ferrosilite + magnetite + coesite; **b** $fO_2$ vs. pH diagram at 3 GPa and 700 °C generated by thermodynamic modelling (Deep Earth Water model[26,28]) of aqueous fluids in equilibrium with aragonite-only (blue dot), graphite-only (grey dot) and aragonite + graphite (magenta dot). Black solid lines: saturation curves. Coloured dots: experimental conditions at $fO_2$ buffered by ferrosilite + magnetite + coesite. Grey field: $CO_{2(aq)}$ is the dominant carbon-bearing species; it is adjacent to Ca(HCO₃)+-dominated field at higher pH and to CH₄(aq)-dominated field at lower $fO_2$ values. Neutral pH is shown for reference with a dashed line. Calculations are performed at 3 GPa, 700 °C and $fH_2$ buffered by ferrosilite + magnetite + coesite + H₂O (equivalent to log ($fO_2$/1 bar)= −13.36; $\Delta FMQ = +0.61$).

model[26,28] (Fig. 2b; Supplementary Table 3). Calculations indicate that fluids in equilibrium with pure aragonite should display a basic pH = 5.09 (neutral pH = 3.09 at 3 GPa and 700 °C) and $XCO_2 = 0.0002$, which is even lower than the measured value. However, the model predicts that the dominant carbon-bearing dissolution product of aragonite at the investigated conditions is the calcium-bicarbonate ion Ca(HCO₃)⁺, in agreement with previous studies[13,14]. According to the model, a substantial concentration of carbon is present in the form of ionic species. In particular, Ca(HCO₃)⁺ accounts for the 74.3 mol% of the total carbon-bearing dissolved species, while $CO_{2(aq)}$ only 7.3 mol% (Supplementary Table 3). The solubility of aragonite is therefore higher than that inferred on the basis of the measured $CO_2$ only. As Ca(HCO₃)⁺ is not measurable by QMS, we rely on the predicted Ca(HCO₃)⁺ abundance to correct up the bulk aragonite solubility to 0.459 mol $CO_2$/kg H₂O, equivalent to $5.28 \times 10^3$ ppm (=mg C/kg solution; 495 ppm of which deriving from measured $CO_{2(aq)}$), which agrees well with previous estimates[14]. In aragonite-only runs, the evolved $CO_2$ is independent of $fO_2$, as demonstrated by the nearly identical $XCO_2 = 0.002$ in the more oxidized ($\approx \Delta FMQ + 2$) control run buffered by Re–ReO₂ (RRO in Fig. 2a; Supplementary Tables 1 and 2). On the contrary, the addition of ~0.5 molal to ~1 molal chlorine to lower pH in control experiments effectively boosts fluid $CO_2$ concentrations to 2.17–2.46 mol $CO_2$/kg H₂O (corresponding to $XCO_2 = 0.038$–0.042; Supplementary Tables 1 and 2).

Compared to runs with aragonite-only, the $CO_2$ amount in fluids in experiments with graphite-only is significantly higher, with $XCO_2 = 0.339$ after 240 h (Fig. 2a) corresponding to 28.5 mol $CO_2$/kg H₂O and a carbon concentration of $1.52 \times 10^5$ ppm, two orders of magnitude higher than in fluids dissolving aragonite-only (Fig. 2a; Supplementary Tables 1 and 2). DEW model calculations show that fluids in equilibrium with pure graphite should display an acidic pH = 2.21 (Fig. 2b) and $XCO_2 = 0.305$ (Supplementary Table 3), which is nearly identical to the value of $XCO_2 = 0.303$ predicted by conventional modelling of graphite-saturated COH fluids[29,30] (Fig. 2a; Supplementary Table 4) and close to the measured value of 0.339. In fluids in equilibrium with graphite-only, $CO_{2(aq)}$ is predicted to account for 99.93% of the total dissolved carbon (Supplementary Table 3).

The $CO_2$ amount measured in time-resolved runs containing mixed aragonite and graphite (Fig. 2a; Tables 1 and 2) ranges from $XCO_2 = 0.026$ after 0.24 h to 0.166 after 240 h. Results suggest that chemical equilibrium is almost reached after 72 h. After 240 h, fluids contain 11.19 mol $CO_2$/kg$_{water}$ corresponding to $8.96 \times 10^4$ ppm C, about one half that in fluids in equilibrium with graphite-only. This decline is not anticipated by either the DEW model calculations–which predict only a slightly lower $XCO_2$ of 0.284 for fluids in equilibrium with graphite-only (Supplementary Table 3)—or by the graphite-saturated COH fluid model—which cannot account for the dissolution of aragonite because it does not consider components other than C, O and H. Our experimental result confirms that the estimation of $XCO_2$ with available thermodynamic models is hampered in complex systems[15]. Nevertheless the DEW model is still useful to predict that fluids in equilibrium with both aragonite and graphite display a basic pH = 4.2 (Supplementary Table 3), which is higher than fluids in equilibrium with graphite-only but lower than fluids in equilibrium with aragonite-only, dominated by the dissolution product Ca(HCO₃)⁺ (Fig. 2b). The consequence is that, as in fluids in equilibrium with graphite alone, $CO_{2(aq)}$ is calculated to predominate in fluids in equilibrium with both graphite and aragonite (Fig. 2b), with $CO_{2(aq)}$ accounting for 91.4% of the total carbon-bearing species, whereas Ca(HCO₃)⁺ constitutes only 3.70% of the total carbon in solution (Supplementary Table 3).

**Carbon isotopic composition of graphite, aragonite and CO₂.** In experiments where water interacts with single minerals, the isotopic abundances of evolved $CO_2$ are identical to that of the starting materials within analytical uncertainties. In particular, due to the analytical sensitivity of QMS, this is evident in runs characterized by higher $CO_2$ production, i.e. ¹³C-aragonite + chlorine and ¹²C-graphite-only runs. In graphite-alone runs, the average $CO_2$ isotopic abundance is 1.11% ¹³C after 240 h (Supplementary Table 1), indistinguishable from the ¹³C abundance of the starting graphite (1.09% ¹³C measured by IRMS; Supplementary Fig. 2) considering analytical uncertainties.

The isotopic signature of evolved $CO_2$ in experiments with both ¹³C-aragonite and ¹²C-graphite (Fig. 3a; Supplementary

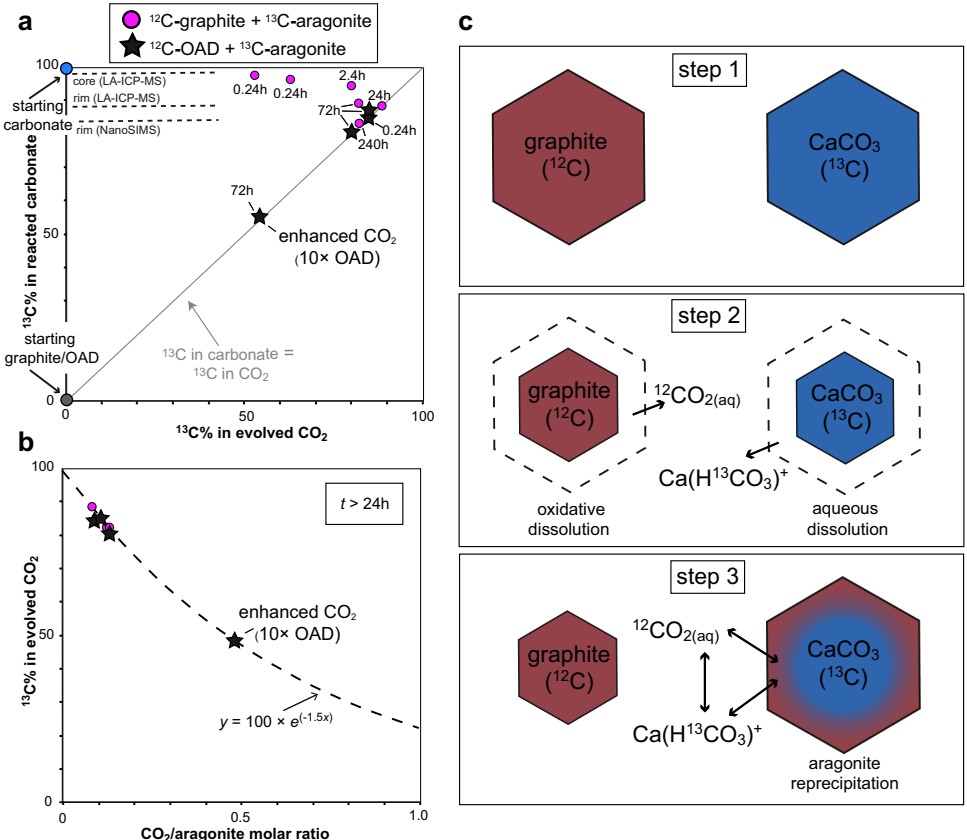

**Fig. 3 Isotopic composition of experimental fluids and solids. a** Measured $^{13}C$ abundance (%) in $CO_2$ versus in bulk after-run aragonite. Grey reference line indicates identical $^{13}C$ abundances in $CO_2$ and in bulk aragonite. Starting $^{13}C$% of carbonate and graphite are indicated with blue and grey dots, respectively. Representative point analyses of aragonite cores and rims are shown with black dashed lines. Magenta dots: $^{13}C$-$CaCO_3$ + graphite runs. Black stars: $^{13}C$-$CaCO_3$ + oxalic acid di-hydrate (OAD; starting $^{13}C$% is coincident with starting graphite) runs. **b** $CO_2$/$^{13}C$-$CaCO_3$ molar ratio plotted against $^{13}C$ abundance (%) in $CO_2$ of runs with duration >24 h. Dashed line represents the best fit of the data using an exponential equation. **c** Conceptual model for the carbon isotope exchange observed in experiments. At step 1, isotopically pure $^{12}C$-graphite (brown) and $^{13}C$-aragonite (blue) coexist in the same assemblage. As the interaction with water starts (step 2), graphite undergoes oxidative dissolution forming $^{12}CO_{2(aq)}$, while aragonite dissolves forming $Ca(H^{13}CO_3)^+$, as predicted by thermodynamic modelling. While graphite oxidative dissolution is self-limiting because the maximum amount of $CO_{2(aq)}$ in the fluid is constrained by the redox state of the system, aragonite undergoes a continuous process of dissolution/precipitation (step 3). Therefore, the final isotopic composition of $CO_{2(aq)}$, which is in dynamic equilibrium with $Ca(HCO_3)^+$ and thus with aragonite, becomes rapidly enriched in $^{13}C$ ending with $^{13}C$ abundances of >80% of the starting $^{13}C$-aragonite.

Table 1) is relatively rich in $^{12}C$ ($^{13}C$ = 52.9–62.8%) only in runs characterized by a very short duration of 0.24 h, while it becomes relatively homogeneous for $t \geq 24$ h reaching $^{13}C$ abundance = 82.2% after 240 h. Bulk measurements by means of total carbonate HCl dissolution allowed retrieval of the final average isotopic composition of after-run aragonite coexisting with $CO_2$ (y-axis in Fig. 3a; Supplementary Table 1). Low $^{13}C$ abundances in evolved $CO_2$ are always coupled with high $^{13}C$ in bulk aragonite, the latter tending to the starting $^{13}C$-$CaCO_3$ value of 99.4% $^{13}C$ (blue dot in Fig. 3a) in the shortest 0.24 h ($^{13}C$ = 96.1–97.2%) and 2.4 h ($^{13}C$ = 95.8%) runs (Fig. 3a; Supplementary Table 1). In longer $t \geq 24$ h runs, $^{13}C$ abundances of $CO_2$ and of bulk carbonate converge (cf. black dashed line in Fig. 3a). After 240 h, the $^{13}C$ abundance in aragonite is 82.9%, nearly identical to that of evolved $CO_2$ (i.e. 82.2%).

Micro-analyses performed by means of NanoSIMS and LA-ICP-MS (dashed lines in Fig. 3a; Supplementary Tables 5 and 6) show that aragonite crystals are isotopically zoned, in particular in short runs, with recrystallized rims characterized by $^{13}C$ abundances comparable to that of bulk aragonite and of $CO_2$ in longer runs (e.g. $^{13}C$ = 81.6(8)% in the 24 h run COH106; Supplementary Table 6), and relict cores showing high $^{13}C$

abundance up to 98(1)% $^{13}C$ (e.g. run COH97; Supplementary Table 6) approaching the starting composition of $^{13}C$-$CaCO_3$. Conversely, the $^{13}C$ abundance in after-run graphite indicates negligible isotopic re-equilibration (Supplementary Fig. 2; Supplementary Table 5, 6). NanoSIMS measurements range from 1.132(6)% to 1.195(3)% (Supplementary Fig. 2; Supplementary Table 5), concordant with LA-ICP-MS measurements ranging from $^{13}C$ = 1.1(1)% to 1.5(2)% (Supplementary Fig. 2; Supplementary Table 6).

The $^{12}C$-rich composition of $CO_2$ in the shortest runs and the lack of newly formed graphite suggest that in mixed aragonite–graphite runs, the early source of $CO_2$ was graphite undergoing irreversible oxidation. To validate this hypothesis, we performed a series of control experiments at identical experimental conditions but without graphite, where the early source of $^{12}CO_2$ is provided by oxalic acid di-hydrate ($^{13}C$ = 1.29%; OAD and black stars in Fig. 3a), which decomposes already at $T \approx 200$ °C to a mixed $H_2O$–$CO_2$ fluid[31]. OAD has been added in the proper amount to keep the same $CO_2$/$^{13}C$-$CaCO_3$ ratio $\approx$ 0.1 characterizing graphite + aragonite experiments (Fig. 3b; Supplementary Table 1). In addition, 10 times more OAD was added in a single run to increase the $CO_2$/$^{13}C$-$CaCO_3$ ratio to a

value of ~0.5 ("enhanced $CO_2$" in Fig. 3a, b; Supplementary Table 1). In OAD + $^{13}C$-$CaCO_3$ runs with $CO_2$/$^{13}C$-$CaCO_3$ ratio ≈ 0.1, the isotopic composition of both $CO_2$ and bulk carbonate shows no appreciable differences compared to aragonite + graphite runs, further supporting the hypothesis of graphite as the early source of $CO_2$. In addition, the OAD + $^{13}C$-$CaCO_3$ run with $CO_2$/$^{13}C$-$CaCO_3$ ratio ≈ 0.5 makes evident that the $^{13}C$ abundance of $CO_2$ (and of coexisting aragonite) varies as a function of the $CO_2$/$^{13}C$-aragonite ratio (CAR), which can be described by the following exponential function (Fig. 3b):

$$^{13}C_{CO_2}\% = 100 \times e^{(-1.5 \times CAR)} \quad (1)$$

which represents a mathematical model for the competing isotopic buffering in fluids where $CO_2$ originated from the oxidative dissolution of $^{12}C$-graphite interacts with $^{13}C$-aragonite. Equation 1 shows that graphite-derived $CO_2$ (and coexisting aragonite) are fully isotopically buffered by $^{13}C$-aragonite to $^{13}C$ abundances ≈ 100% when CAR tends to zero. Conversely, by increasing CAR the contribution of $^{12}C$-graphite to the isotopic composition of $CO_2$ becomes increasingly important, with $^{13}C$ abundances tending to zero (i.e. fully buffered by graphite) at very high $CO_2$/$^{13}C$-aragonite ratios (e.g. $^{13}C_{CO2}$ = 0.01 when CAR = 6).

**A conceptual model for carbon isotopic exchange among graphite, aragonite and $CO_2$.** In runs containing mixed $^{13}C$-aragonite and $^{12}C$-graphite, the isotopic composition of $CO_2$ after 24 h is nearly coincident with that of the recrystallized aragonite, although it is produced mainly by the oxidative dissolution of graphite which remains isotopically unchanged. Our experimental results confirm that carbon isotope exchange between graphite and aragonite at 700 °C is sluggish, in agreement with previous findings suggesting that carbon diffusion in graphite is very slow at $T < 1300$ °C [32,33]. In the shortest 0.24 h and 2.4 h runs, despite the similar $^{13}C$ abundances of $CO_2$ and aragonite rims, the bulk carbonate $^{13}C$ is markedly higher because of a number of unreacted $^{13}C$-rich cores persist, as shown by NanoSIMS and LA-ICP-MS analyses. In order to develop a conceptual model for the isotopic exchange among aragonite, graphite and $CO_2$, we rely on observed microtextures, measurements and thermodynamic modelling results suggesting that during the run both graphite and aragonite undergo dissolution. However, $^{12}C$-graphite produces mainly $^{12}CO_2$ by irreversible oxidation, which can be expressed by the following $fO_2$-dependent reaction:

$$^{12}C + O_2 = >^{12}CO_2 \quad (2)$$

$^{13}C$-aragonite dissolves initially forming mainly calcium-bicarbonate ions ($Ca(H^{13}CO_3)^+$; Fig. 3c), according to the reactions involving the predominant species:

$$Ca^{13}CO_3 + H_2O< = >Ca(H^{13}CO_3)^+ + OH^- \quad (3)$$

when the fluid becomes saturated with respect to graphite, $CO_2$ production by graphite oxidation stops and the unreacted graphite remains chemically and isotopically inert, while aragonite continuously dissolves and reprecipitates in dynamic equilibrium with the dissolved aqueous carbon species $CO_2$ and $Ca(HCO_3)^+$, which in turn exchange carbon isotopes according to the following reaction:

$$^{12}CO_2 + Ca(H^{13}CO_3)^+< = >^{13}CO_2 + Ca(H^{12}CO_3)^+ \quad (4)$$

therefore, because of the continuous dissolution/reprecipitation process controlled by Eq. 3, the carbonate is effective in buffering the isotopic signature of $CO_2$ even after a very short time, despite the fact that $CO_2$ is produced mainly by irreversible oxidation of graphite. Equation 1, however, predicts that aragonite buffering is

possible only for relatively low $CO_2$/aragonite ratios, ideally below 0.46 where the $^{13}C$ abundances would be >50%. We will show below that Eq. 1 can be used to evaluate how the isotopic compositions of $CO_2$ evolved from the sedimentary slab at subarc conditions can meet the global average arc signature according to those variables that control the $CO_2$ release, i.e. the fluid/rock ratios and the redox conditions imposed by the environment.

**A mathematical model for δ$^{13}C$ of $CO_2$ produced in the sedimentary slab and comparison with global average arcs.** Thermodynamic considerations require that at graphite saturation and at fixed $P$–$T$ conditions the amount of $CO_2$ produced by oxidation of graphite in a pure, carbonate-free C–O–H system depends solely on: (i) redox conditions—the more oxidizing they are, the higher the $XCO_2$ (=$CO_{2,aq}$/($H_2O+CO_{2,aq}$))[34], and (ii) the amount (number of moles) of $H_2O$ interacting with graphite—the higher it is, the higher the amount of $CO_2$ at fixed $XCO_2$[30,35]. Therefore, at graphite saturation conditions and as long as graphite is present, the amount of $CO_2$ produced by oxidation is not dependent on the amount of graphite in the system.

In graphite + aragonite + $H_2O$ systems, we demonstrated that the carbon contribution to fluids due to aragonite dissolution is negligible at subarc $P$–$T$ conditions compared to graphite, so the amount of $CO_2$ evolved still depends on (i) and (ii) above, with the additional consequence that the graphite/carbonate ratio is again not relevant. Moreover, the carbon content in these fluids is basically ascribable to their $CO_2$ content, as $Ca(HCO_3)^+$ is limited to about 1%. However, we noticed a difference in the aragonite + graphite system compared to the graphite-alone (COH) system: the halving of measured $XCO_2$ (=0.166) compared to graphite-saturated COH fluid models (=0.303; Fig. 2a). By analogy with other systems, we suggest that this is likely related to a modification of water activity[15], in this case due to aragonite dissolution. Applying the experimentally derived correction factor of 0.548 (=0.166/0.303), we calculated for the aragonite + graphite system the $XCO_2$ values predicted by conventional graphite-saturated fluid thermodynamic model[29,30] at different redox conditions (ΔFMQ). The calculated $XCO_2$ were used to calculate the absolute amount of $CO_2$ produced by oxidation of graphite, obtained by fixing the absolute amount (moles) of $H_2O$ in the system. Assuming 1 mol aragonite in the system, fixed $H_2O$ values correspond also to water/aragonite molar ratios (WAR) and $CO_2$ values to $CO_2$/aragonite molar ratios (CAR). Using these last values and Eq. 1, we calculated the $^{13}C$ abundance of $CO_2$ produced by oxidation of $^{12}C$-graphite after interaction with $^{13}C$-aragonite. Because in our experimental model $^{12}C$-graphite and $^{13}CaCO_3$ represent analogues of, respectively, ocean organic matter with δ$^{13}C$ ≈ −20 and marine carbonates with δ$^{13}C$ ≈ 0, we are now able to translate $^{13}C$ abundances (ranging from 0 to 100%) to equivalent δ$^{13}C$ values (ranging from 0‰ to −20‰) using a conventional linear model:

$$\delta^{13}C‰ = 0.2 \times {}^{13}C\% - 20 \quad (5)$$

the calculated δ$^{13}C$‰ of $CO_2$ repeated for 10,000 different random combinations of the parameters ΔFMQ and WAR have been fitted with a three-variate parametric equation to provide, with an estimated average uncertainty of ±0.2‰, the following mathematical model (Fig. 4):

$$\delta^{13}C_{CO_2}‰ = -20 + 20\frac{1}{1 + e^{-f(\Delta FMQ, WAR)}} \quad (6)$$

where the function $f(\Delta FMQ, WAR)$ is a third-order polynomial found to minimize residuals (parameters and associated uncertainties available in Supplementary Table 7). The global arc δ$^{13}C$ ranging from −2.8 to −3.3 ‰[4] (orange box in Fig. 4) intersects the surface of Eq. 6 providing a continuous layer that can be averaged

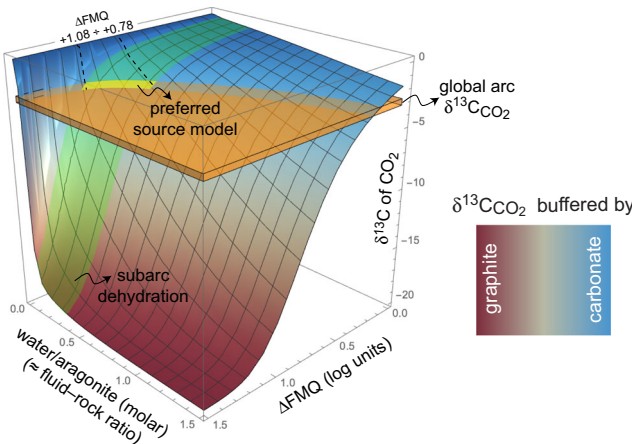

**Fig. 4 Mathematical model showing the predicted δ¹³C of CO₂ as a function of both ∆FMQ and water–aragonite (≈ fluid–rock) molar ratio.**
Modelled CO₂ is originated from oxidative dissolution of graphite and interacts with aragonite and aqueous fluids at 3 GPa and 700 °C. Brown surface color: $\delta^{13}C$ buffered by graphite. Blue surface color: $\delta^{13}C$ buffered by aragonite. Orange box: global average $\delta^{13}C$ of arc CO₂[4]; the intersection with the mathematical model shows that arc CO₂ signatures are met for a wide range of ∆FMQ and fluid–rock ratios. Preferred model (yellow field) corresponds to the intersection among our mathematical model, the global average arc $\delta^{13}C$ and typical fluid–rock ratios at slab-top conditions assuming percolating flux (green field; 0.05–0.33 molar = 0.01–0.06 mass). In this case, ∆FMQ ranging from +0.78 to +1.08 are derived for the source of carbon in the sedimentary slab at subarc conditions.

and expressed with the following equation:

$$\Delta FMQ = \frac{2}{1 + e^{(A + B \times WAR + C \times WAR^2 + D \times WAR^3)}} \qquad (7)$$

equation 7, where $A = -0.314(9)$, $B = 2.75(5)$, $C = -1.46(6)$ and $D = 0.37(2)$, represents the locus of points of all the possible combinations of ∆FMQ and water/aragonite ratio that result in the isotopic composition of sediment-derived CO₂ satisfying the global average arc values.

**Comparison with conventional mass-balance/equilibrium fractionation models.** The observed carbon isotopic exchange among CO₂, graphite and aragonite is potential of interest for subducted sediments buried in a wide range of basins; the only limiting factor is the coexistence of calcium carbonate with elemental carbon (formerly organic matter) in whatever proportion and the presence of an aqueous fluid. Our experiments at 3 GPa and 700 °C show that CO₂ produced by oxidative dissolution of ¹²C-graphite ($^{13}C = 1.1\%$; equivalent $\delta^{13}C = -19.8$, cf. Eq. 5 and Supplementary Table 1) becomes rapidly enriched in ¹³C ending after 240 h with $^{13}C = 82.2\%$ (equivalent $\delta^{13}C = -3.56$) due to its isotopic exchange with ¹³C-aragonite ($^{13}C = 99.6\%$; equivalent $\delta^{13}C = -0.08$), despite the overwhelming abundance of graphite in the starting materials (graphite/aragonite$_{(molar)} = 5.6$; see below).

Equivalent $\delta^{13}C$ values can be compared with conventional models combining mass-balance calculations with isotope equilibrium fractionation (see Methods). Let's assume a mixture similar to what investigated experimentally: aragonite ($\delta^{13}C = 0‰$, by analogy with a marine carbonate source) + graphite ($\delta^{13}C = -20‰$; by analogy with a marine organic matter source), equilibrating at 700 °C (pressure effect ignored). Assuming complete solid–solid isotopic exchange and neglecting devolatilization, the final isotopic composition would be a function of the

molar ratio between graphite and aragonite, for instance: 1) $\delta^{13}C_{graphite} = -5.29‰$; $\delta^{13}C_{aragonite} = -0.14‰$ for graphite/aragonite = 0.01; 2) $\delta^{13}C_{graphite} = -12.57‰$; $\delta^{13}C_{aragonite} = -7.43‰$ for graphite/aragonite = 1.0; 3) $\delta^{13}C_{graphite} = -17.77‰$; $\delta^{13}C_{aragonite} = -12.63‰$ for graphite/aragonite = 5.6. With the latter abundance value, comparable with the experimental setup, calculations show that the isotopic composition of all the phases in equilibrium would vary slightly as a function of the amount of CO₂ in the system produced by oxidation of graphite (Supplementary Fig. 3a), with $\delta^{13}C_{CO2} = -10.17‰$ for CO₂/aragonite = 0 and $\delta^{13}C_{CO2} = -11.33‰$ for CO₂/aragonite = 1. This conventional model is therefore not adequate to reproduce our experimental system, characterized by CO₂/aragonite ≈ 0.1, where both aragonite and CO₂ converge to much heavier compositions. However, the fit between the isotopic model and the experimental trend improves significantly (Supplementary Fig. 3a) if we assume (1) graphite as chemically reactive but isotopically inert phase and (2) starting CO₂ displaying $\delta^{13}C = -20‰$, i.e. the same value of the starting graphite. These assumptions, deduced on the basis of our experimental findings, underline again the need for experimental constraints on deep isotopic exchange processes involving fluids. For instance, an implication of our study could throw some light on the hard debate on the genesis of diamonds characterized by light, organic matter-like isotopic compositions[19]. In fact, even if we predict that, after having interacted at subarc conditions with graphite-saturated fluids, sedimentary carbonates will be only slightly depleted in ¹³C, showing the same $\delta^{13}C$ −2.8 to −3.3‰ characterizing CO₂, they could become locally lighter (cf. brown in Fig. 4) in case of highly oxidized conditions[36] or very high water/carbonate (≈fluid/rock) ratios[37]. Because of their stability at high-pressure conditions[38], these carbonates could be subducted further, eventually making available a ¹²C-enriched source of carbon, which comes from carbonate and not organic matter, in the diamond stability field[39] (Fig. 5).

**The link between sedimentary slab fluids and arc emissions.** As an aqueous fluid-phase species, CO₂ originated from subducted sediments close to the slab-mantle interface is particularly effective in metasomatizing the subarc supra-subduction mantle, prompting carbonation reactions[20,21,23,40] that may facilitate diapiric upwelling and melting processes[20,41]. Therefore, there is good reason to believe that sediment-derived CO₂ can impose its isotopic signature on the subarc mantle, the source of arc magmatism (Fig. 5). The interaction of subducted ocean carbonates and organic matter with fluids coming from the dehydration of the down-going slab[25,42] is a continuous process starting early in the forearc region[9,43] (Fig. 5; Supplementary Fig. 1). However, our study indicates that the production of CO₂ is strictly connected to the stability of graphite in aqueous fluids. At forearc conditions graphite is poorly soluble by oxidative dissolution, even considering the increased dissolution susceptibility for disordered graphitic carbon, in the absence of intense flushing by aqueous fluids (i.e. channelized flow)[16,35]. Aqueous fluids reach their maximum CO₂ contents at subarc conditions (Fig. 5; Supplementary Fig. 1), which are the conditions investigated in our study[16]. Beneath arcs, the interaction between aqueous fluids coming from the dehydrating oceanic slab and percolating into the overlying carbonate metasediments will thus result in a large amount of CO₂ leached out from the slab, as long as graphite is not completely removed by oxidative dissolution, which depends chiefly on the time-integrated fluid flux for given redox state[18]. Although channelized fluid flow can potentially dissolve the entire amount of organic carbon at blueschist-to-eclogite conditions, the geological record suggests the complete removal is difficult even for very high fluxes and often limited to the more

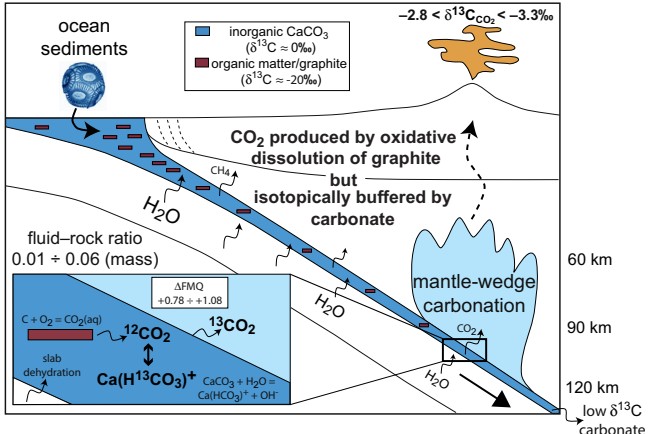

**Fig. 5 Carbon isotope exchange at the subarc slab-mantle interface.**
Organic (brown) and inorganic (dark blue) carbon occurring in open-ocean sediments enters the subduction channel, where it interacts with aqueous fluids produced by dehydration of the down-going oceanic lithosphere. While calcium carbonate dissolution increases almost linearly with depth[14], resulting in fluids containing ~5000 ppm C mainly as $Ca(HCO_3)^+$ at subarc depth, the oxidative dissolution of graphite increases exponentially and reaches its maximum at subarc conditions, where fluids coexisting with graphite contain ~140,000 ppm C mainly under the form of $CO_{2(aq)}$. With the simultaneous presence of aragonite and graphite, experiments show that the carbon concentration in aqueous fluids is halved, but still controlled by oxidation of graphite to $CO_2$. Nevertheless, as aragonite is very prone to dissolution-reprecipitation processes, it shows an outstanding capacity to buffer the isotopic signature of graphite-derived $CO_2$ to carbonate-like values characterizing the global arc $CO_2$ (orange field). In order to exert its buffering capacity, graphite-derived $CO_2$ must be low with respect to carbonate, which is the case for low fluid–rock ratios coupled with oxidizing conditions (preferred model) or for high fluid–rock ratios coupled with relatively reducing conditions. We suggest that, as a mobile but reactive fluid-phase, $CO_2$ is particularly effective in metasomatizing the supra-subduction mantle (light blue field), eventually imposing its isotopic signature to the deep source of arc magmatism.

soluble disordered forms of organic carbon[18]. Fully crystalline graphite, which is characterized by sluggish rates of isotopic diffusion[3], is thus expected to persist at subarc depths. We demonstrated experimentally that the organic carbon content is not a relevant variable in controlling the carbon isotopic signature of $CO_2$ produced by dissolution processes, which only depends on redox conditions and fluid-rock ratios. Therefore, $\delta^{13}C$ of gaseous arc emissions is not a simple linear combination of the amount of buried organic and inorganic carbon[4]. One major implication is that fluctuations in $\delta^{13}C_{CO_2}$ do not necessarily reflect variations in the source input, for instance, due to shallow crustal carbonate assimilation[44,45] or to the burial of anoxic sediments dominated by organic matter[19]. It could reflect instead local variations in fluid-rock ratios (i.e. channelized vs. percolating flux) or in the redox state (i.e. oxidizing vs reducing conditions). Equation 7 can be used to constrain the average redox state (expressed as ΔFMQ) in the sedimentary carbon source of the slab that meets the global arc $\delta^{13}C_{CO_2}$. ΔFMQ values ranging from +0.78 to +1.08 are predicted assuming a fluid–rock mass ratio (≈WAR) of 0.01–0.06 (=0.05–0.33 molar), which has been suggested for infiltrating, non-channelized fluids produced by dehydration reactions in eclogites[46,47]. These oxygen fugacity values are in agreement with estimations of island-arc-basalt (IAB) sources[48,49] and supra-subduction mantle peridotites equilibrated at subarc depths[50]. It remains an open question how

these oxygen fugacity values correlate with the amount of oxygen[36] transferred from the slab to the IAB source, with consequences for the redox budget of subarc mantle[51].

## Method

**Experimental approach and characterization of the solid- and fluid phases**. In this study, we employed as starting materials: (i) labelled $Ca^{13}CO_3$ (calcite; forming aragonite at run conditions) (Sigma–Aldrich); (ii) synthetic graphite (Sigma–Aldrich), highly ordered as suggested by Raman spectroscopy[16]; (iii) oxalic acid di-hydrate (Sigma); (iv) MilliQ water, boiled while flushed with $N_2$ to remove dissolved atmospheric $CO_2$. Experiments were buffered using the double-capsule technique[52] (Supplementary Fig. 4) with an inner $H_2$-permeable $Au_{60}Pd_{40}$ capsule, containing the starting materials with the addition of water (~20 wt%), and an outer Au capsule filled with the buffering assemblage fayalite + magnetite + quartz + $H_2O$ (FMQ; forming ferrosilite + magnetite + coesite at run conditions, verified by electron microscopy, electron microprobe analysis and Raman spectroscopy; Supplementary Fig. 5). The buffer in the outer capsule constrains directly the $fH_2$ in the inner capsule. The redox conditions ($fO_2$) in the inner capsules are constrained indirectly by the buffer, whose ΔFMQ is +0.76 log units in the outer capsule, to slightly lower $fO_2$ values (ΔFMQ = +0.61 log units; see "Thermodynamic modelling" and Supplementary Table 4).

Experiments were performed at 3 GPa and 700 °C using an end-loaded piston-cylinder apparatus. Temperatures were measured with K-type thermocouples and considered accurate to ±5 °C. Pressure calibration is based on the quartz/coesite transition[53] and accurate to ±0.01 GPa. Runs with $CaCO_3$ were first pressurized at 3 GPa for 2 h, to promote carbonate crystal growth by cold sintering[54] in order to increase the grain size of the synthetic micrometric $CaCO_3$ power. Samples are then heated to 700 °C with a ramp of 100 °C/min. Run durations were from 2.4 to 240 h. Quench is obtained by cutting off the power supply, resulting in a temperature decline of >40 °C per second. $CO_2$, water and other volatiles in the inner capsules were measured quantitatively by quadrupole mass spectrometry (QMS) using the capsule-piercing technique[31]. The typical analytical uncertainty is 1 mol% for $CO_2$ and $H_2O$. Solid phases were checked by scanning electron microscopy, electron microprobe analyses and micro-Raman spectroscopy.

**Isotopic analysis of $CO_2$**. The quantitative analysis and the determination of the $^{13}C/^{12}C$ ratio of $CO_2$ have been performed simultaneously by means of QMS[31], monitoring the $m/z$ channel 45 ($^{13}CO_2$) in addition to channels considered for routine analysis. The calibration curve linking the $^{13}C/^{12}C$ ratio of $CO_2$ and the ratio of the integrated peaks of channels 45 ($^{13}CO_2$) and 44 ($^{12}CO_2$) (44/45 ratio) has been derived by measuring 3 different mixtures with known $^{13}C/^{12}C$ ratio prepared to start from (i) regular oxalic acid di-hydrate (OAD; Sigma–Aldrich), used also for OAD + $Ca^{13}CO_3$ experiments and (ii) isotopically nearly pure $^{13}C$ oxalic acid di-hydrate (Sigma–Aldrich), thermally decomposed to $CO_2$ at 250 °C[31] (Supplementary Table 8). Analyses performed by QMS are affected by a very small mass-bias effect. Actually, the regression line provided a very low correction factor of 1.03 for the integrated peak of channel 45.

**Isotopic analysis of solids: NanoSIMS**. $^{13}C/^{12}C$ ratios of carbonate and graphite grains in the experimental sample were determined by nanoscale secondary ion mass spectrometry (NanoSIMS). Measurements were conducted on the Cameca NanoSIMS 50 installed at Muséum National d'Histoire Naturelle of Paris. The sample capsule was included in pure indium and gold-coated (20 nm thick). Secondary ions of $^{12}C^{12}C^-$ and $^{13}C^{12}C^-$ were collected in multicollection mode to obtain $^{13}C/^{12}C$ ratios after calibrating with natural abundance graphite and calcite samples. $^{16}O^-$ secondary ions were used to locate graphite versus carbonate grains. Mass resolving power was set at a minimum 10,000, enough to resolve interferences on measured secondary ions. Before each analysis, a $5 \times 5$ μm² surface area was initially pre-sputtered for 60 s with a 500 pA primary $Cs^+$ rastering beam, in order to remove the gold coating and reach a sputtering steady-state[55]. For analyses of graphite grains, the primary beam was set to 1 pA and is was scanned over a surface area of $5 \times 5$ μm². Nevertheless, to avoid surface contamination, only ions from the inner $2.6 \times 2.6$ μm² regions were collected with the "beam blanking mode". Each analysis consisted of a stack of 100 cycles, with a duration of 2.048 s each. Similar settings were used to carbonate grains, except that the primary beam was set to 9 pA and each cycle was 8.192 s long. $^{13}C/^{12}C$ of carbonate grains were also investigated using $^{12}C^-$ and $^{13}C^-$ secondary ions (mass resolving power set to 9000), showing no significant deviation from the measurements using $C_2^-$ secondary ions. Due to large isotopic variations observed in these samples, instrumental mass fractionation (of a few per mil) was neglected.

**Isotopic analysis of solids: LA-ICP-MS**. $^{13}C/^{12}C$ ratios of carbonate and graphite grains in the experimental sample were also determined by LA-ICP-MS. Measurements were carried out over a New Wave UP 266 laser ablation system coupled with a Thermo Fisher ICAP-Q ICP-MS (University of Insubria). In order to find the best compromise between ablated surface and sensitivity, ablation conditions were optimized over a natural abundance calcite sample with a known $^{12}C/^{13}C$ ratio (see Section "Isotopic analysis of solids: Bulk analyses"). As a result, for each

spot determination, 20 shots within 1 s were performed, using a 30 μm diameter circular spot size and adjusting the laser power to obtain a fluence value of around 17 J/cm[2]. Helium was used as the carrier gas (0.85 L/min). Under these conditions the hole depth (subsequently evaluated by scanning electron microscopy) is, on average, about 20 μm. Analytical accuracies and uncertainties have been evaluated by measuring the following internal standards: (i) natural abundance calcite, characterized isotopically by GasBench isotope ratio mass spectometer (IRMS) in two different laboratories (Florence and Milan), displaying an average $^{13}$C abundance of 1.1208(2) %; (ii) synthetic labelled $CaCO_3$ produced by precipitation from a $Na_2^{13}CO_3 + Na_2^{12}CO_3$ solution treated with $CaCl_2$, with $^{13}$C abundance of 43.8% measured by HCl dissolution followed by QMS analysis of evolved $CO_2$ (see "Bulk analyses" below). LA-ICP-MS measurements on these standards have been proven to be accurate to ~2% for natural calcite (standard deviation 5.8%) and ~1% for synthetic $^{13}C-^{12}C$ calcite (standard deviation 0.7%) (Supplementary Table 9). Analyses of graphite are affected by a higher standard deviation of about 12%, due to decreased ablation efficiency.

**Isotopic analysis of solids: Bulk analyses.** Bulk analyses of the carbonates contained in the experimental capsules have been carried out by HCl decomposition followed by QMS analysis of evolved $CO_2$ (see "Isotopic analysis of $CO_2$"). To perform this analysis sample capsules coming from LA-ICP-MS determination were placed in a U-shaped glass tube located upstream with respect to the QMS analyser. After 5 outgassing cycles with Ar, 1 mL of HCl (4.5 M) was introduced (from an additional port) in the bottom of the U-tube, where the capsule is located. The tube was then closed (by means of a bypass valve of the gas manifold) for 30 min in order to allow the complete decomposition of all carbonates: after this reaction time the tube was then placed in line with the QMS analyser to determine the composition of evolved $CO_2$. The analysis of the natural calcite standard provided a $^{13}$C abundance of 1.13%, which is accurate to 0.8% of the certified value. In addition, this method allowed to retrieve the $^{13}$C abundance of the synthesized $^{13}C-^{12}C$ calcite (43.8%) and of $Ca^{13}CO_3$ used as starting material (99.4%) (Supplementary Table 10).

The isotopic characterization of the graphite used as starting material has been performed using EA-IRMS analysis, which yielded a value of 1.0948(2)% $^{13}$C.

**Thermodynamic modelling of fluid composition.** To retrieve the redox conditions in the double-capsule system, oxygen and hydrogen fugacities have been calculated by conventional thermodynamic modelling (Supplementary Table 4). In the outer capsule, containing the buffering assemblage ferrrosilite + magnetite + coesite + $H_2O$, log ($fO_2$/1 bar) = –13.21 and log ($fH_2$/1 bar) = 2.195 have been calculated at $P–T$ conditions of 3 GPa and 700 °C using the Perple_X package[30], considering the thermodynamic dataset of Holland and Powell[56] revised by the authors in 2004 (hp04ver.dat) and the equation of state "H–O HSMRK/MRK hybrid" of the routine "fluids". Then, $XCO_2$ [=$CO_2$/($H_2O$+$CO_2$)$_{molar}$] = 0.303 for graphite-saturated fluids has been calculated by fixing log ($fH_2$/1 bar) = 2.195, which is homogeneous in the inner and the outer capsule, using the Perple_X equation of state of Connolly and Cesare[29] (cf. refs. [15,16] for other details). This $XCO_2$ corresponds to an inner-capsule log ($fO_2$/1 bar) = –13.36 (ΔFMQ = +0.61 log units), which reflects the redox conditions occurring in the runs bearing graphite. Log ($fH_2$/1 bar) = 2.195 has also been used to calculate the fluid speciation and the pH in our experimental systems, using the Deep Earth Water (DEW) model[26,27] (Supplementary Table 3).

**Conventional isotopic modelling.** The global $\delta^{13}$C of the investigated system can be expressed with the following mass balance:

$$\delta^{13}C_{global} = m_{graphite} \times \delta^{13}C_{graphite\_i} + m_{aragonite} \times \delta^{13}C_{aragonite\_i}, \tag{8}$$

where $m$ represents the molar fractions, and $\delta^{13}C_{graphite\_i}$ and $\delta^{13}C_{aragonite\_i}$ the initial composition of graphite and aragonite, respectively, –20‰ and 0‰ in our isotopic model.

Assuming that $CO_2$ is produced only through oxidation of graphite (graphite_ox), and that equilibrium conditions occur between $CO_2$ and graphite, the isotopic composition of graphite_ox is:

$$\delta^{13}C_{graphite\_ox} = (1-f) \times (\delta^{13}C_{graphite\_i} - \Delta_{CO_2-graphite}) + f \times \delta^{13}C_{graphite\_i} \tag{9}$$

where $\Delta_{CO_2}$–graphite is the $CO_2$–graphite equilibrium fractionation factor at 700 °C[57] and $f$ is the molar fraction of oxidated graphite, i.e. ($m_{graphite} - m_{CO2}$)/$m_{graphite}$, ranging from 1 (no $CO_2$ produced) to 0 (graphite fully oxidated to $CO_2$).

The isotopic composition of $CO_2$ produced by graphite oxidation is:

$$\delta^{13}C_{CO_2} = \delta^{13}C_{graphite\_ox} + \Delta_{CO_2-graphite}, \tag{10}$$

so that the lobal $\delta^{13}$C including $CO_2$ ($\delta^{13}C_{global\_ox}$), which in a closed system is numerically identical to $\delta^{13}C_{global}$, can be expressed as:

$$\delta^{13}C_{global\_ox} = (m_{graphite} - m_{CO_2}) \times \delta^{13}C_{graphite\_ox} + m_{CO2} \times \delta^{13}C_{CO_2} + m_{aragonite} \times \delta^{13}C_{aragonite\_i}, \tag{11}$$

Assuming isotopic equilibrium between $CO_2$, graphite_ox and aragonite (Supplementary Fig. 3a), the final compositions are:

$$\delta^{13}C_{aragonite\_f} = \delta^{13}C_{global\_ox} - (m_{graphite} - m_{CO_2}) \times -\Delta_{aragonite-graphite}) - (m_{CO_2} \times -\Delta_{aragonite-CO_2}); \tag{12}$$

$$\delta^{13}C_{graphite\_f} = \delta^{13}C_{aragonite\_f} - \Delta_{aragonite-graphite}; \tag{13}$$

$$\delta^{13}C_{CO_2\_f} = \delta^{13}C_{aragonite\_f} - \Delta_{aragonite-CO_2}, \tag{14}$$

In calculations considering graphite as isotopically inert (Supplementary Fig. 3b), we assume that $CO_2$ is produced by the oxidation of graphite, but that $CO_2$ does not equilibrate with graphite getting oxidized.

Therefore, only $CO_2$ and aragonite are freely exchanging isotopes. In this case, $m_{graphite}$ becomes 0 and $\delta^{13}C_{CO2}$ displays the constant value of –20‰. Therefore, the following equations were used instead:

$$\delta^{13}C_{global} = m_{CO_2} \times 20‰ + m_{Aragonite} \times \delta^{13}C_{aragonite\_i}; \tag{15}$$

$$\delta^{13}C_{aragonite\_f} = \delta^{13}C_{global} - (m_{CO_2} \times \Delta_{aragonite-CO_2}); \tag{16}$$

$$\delta^{13}C_{CO_2\_f} = \delta^{13}C_{aragonite\_f} - \Delta_{aragonite-CO_2}. \tag{17}$$

## Data availability
The authors declare that the data supporting the findings of this study are available within the article.

## Code availability
The authors declare that the codes developed for this study are available within the article.

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

## Acknowledgements

Elena Ferrari and Chiara Compostella (University of Milan) are acknowledged, respectively, for the IRMS analyses of natural abundance calcite standard, provided by Marilena Moroni (University of Milan), and of synthetic graphite used as starting material. Andrea Amalfa and Francesca Miozzi (University of Milan) helped in preparing some of the experiments. Andrea Risplendente (University of Milan) assisted to perform microprobe analysis and scanning electron microscopy. Luca Toffolo (University of Padova, Italy) and Patrizia Fumagalli (University of Milan) are acknowledged for micro-Raman spectroscopy. L.R. acknowledges the NanoSIMS facility at the Muséum National d'Histoire Naturelle in Paris, established by funds from the CNRS, Région Ile de France, Ministère délégué à l'Enseignement supérieur et à la Recherche, and the Muséum National d'Histoire Naturelle. S.T. and S.P. acknowledge support from the Italian program MIUR PRIN 2017ZE49E7_002. ST acknowledges support of the SEED (Grant RV_PSR_SOE_2020_AVILL) and the APC central funds of the University of Milan. This work is part of a project that has received funding from the European Research Council (ERC) under the European Union's Horizon 2020 research and innovation programme (Grant agreement No. 864045).

## Author contributions

S.T. conceived the experimental study, analyzed the experimental fluids by QMS, wrote the manuscript and is responsible for data reduction and analysis. S.R. and D.S. performed the LA-ICP-MS and the bulk carbonate analysis by QMS. L.R. carried out the NanoSIMS analysis. C.T. prepared most of the experiments. D.A.S. supervised the thermodynamic modelling of fluids. C.E.M., D.A.S. and S.P. collaborate to the final version of the manuscript. A.V.B. and A.B. performed the conventional isotopic modelling. S.P. provided funds to support this study.

## Competing interests

The authors declare no competing interests.

**Additional information**

