## [Peer Review File · Nature Communications]

Subducted organic matter buffered by marine carbonate
rules the carbon isotopic signature of arc emissionsREVIEWER COMMENTS

Reviewer #1 (Remarks to the Author):

Dear Authors,
I have read with great interest your contribution.

I prepared an annotated version of the manuscript with some minor comments and minor comments/suggestions on the text.

I think this is an extremely interesting study that should be published after relatively minor adjustments. Specifically I think already starting from the abstract there should be more emphasis on the impact of your study on our understanding of volcanic gas emissions and the global carbon cycle. For instance the study of Mason et al. (2017) on the base of collected data and on mass balance (which your study seem to show are potentially not appropriate) draw conclusions on the contribution of assimilation of shallow crustal carbonates to volcanic gas emissions and its impact on the delta 13 carbon record over Earth's history. Discussing these aspects in light of your experimental results will increase the impact of your study on the scientific community. As written now, abstracts and the final part of the manuscript do not fully highlight the importance of your study, which I think is an essential requirement especially for publication in Nature Communications.

Your study has also important implications for a very intense discussion ongoing around the impact of crustal carbonate assimilation on the intensity of volcanic eruptions (e.g. Freda et al., 2010; Troll et al., 2012). In particular, the C isotopic signature of volcanic gas emission is used as an evidence for assimilation of upper crustal carbonates, increased CO₂ content of the magmas, which therefore can produce more explosive eruptions. If the source of CO₂ is actually from sediments at the slab interface, this would serve to rediscuss the drivers of large explosive eruptions in locations such as Merapi or Colli Albani. I do not stand on any side (yet), but I think it would be interesting to discuss some implications of your findings also on this theme.

In conclusions, I find this study extremely interesting, but I think the (many) implications of the experimental results should be more clearly explained for the community to fully appreciate the quality of your contribution.

REFERENCES

- Freda, C., Gaeta, M., Giaccio, B., Marra, F., Palladino, D.M., Scarlato, P., Sottili, G., 2010. CO₂-driven large mafic explosive eruptions: the Pozzolane Rosse case study from the Colli Albani Volcanic District (Italy). *Bull. Volcanol.* 73, 241–256.
- Troll, V.R., Hilton, D.R., Jolis, E.M., Chadwick, J.P., Blythe, L.S., Deegan, F.M., Schwarzkopf, L.M., Zimmer, M., 2012. Crustal CO₂ liberation during the 2006 eruption and earthquake events at Merapi volcano, Indonesia. *Geophys. Res. Lett.* 39, n/a-n/a. <https://doi.org/10.1029/2012GL051307>

Reviewer #2 (Remarks to the Author):

Deep carbon cycle process and flux in subduction zones is a hot but controversial research topic. Stable carbon isotope is a powerful tool to trace carbon cycle process in subduction zones, however, the carbon isotopic fractionation behavior between different carbon-bearing phases in subduction zones are still unclear. In this study, Tumiami et al., experimentally investigated the carbon isotopic exchange between aragonite and graphite with water at 3 GPa and 700 °C under oxygen fugacity controlled at $-FMQ + 0.61$. They conclude that the global average isotopic composition of volcanic arc emission is explained by a mixture of aragonite + graphite in subduction zone based on experimental data and thermodynamical modeling which is very important for understanding the deep carbon cycling. The data and method are reliable, and the discussion and conclusion are correct for publication after major revision. However, the following questions need to be considered before it is

accepted for publication.

1. The experiments design is novel. It is very impressed that graphite behaves as chemically reactive but isotopically inert phase in experiments. If this is a kinetic process, I am wondering if we can apply the experimental results to interpret natural process in geological time.
2. For the oxidation reaction of graphite (Eq. 2), We think O₂ is not existing in water-rock system at high pressure in subduction zone, so what is the oxidizing medium for this reaction?
3. For the dissolution of CaCO₃ in H₂O, the thermodynamical modeling show the predominant phase in water is considered as Ca(HCO₃)⁺. We repeated the DEW modeling in CaCO₃ and water system, and found at least the CO₂(aq), H₂CO₃(aq) are in same concentration level as Ca(HCO₃)⁺, they should not be ignored.
4. The global average carbon isotopic composition of volcanic arc emission was summarized ranges from -2.8 to -3.3 ‰ in this study. However, the carbon isotopic composition of volcanic arc emission around Pacific Ocean are totally heterogeneous from west to east due to input different ratio of carbonate and organic carbon (Plank and Manning, 2019). I think authors can apply these data here and discuss the how input carbon source affect carbon isotopic composition of volcanic arc emission.

Some minor comments are below:

Line 75: recycled sedimentary carbonate + mantle carbon = arc magmas?

Line 89: pure graphite can not represent organic matter, there are some hydrocarbon compound in organic matter, which may stabilized at 3 GPa and 700 oC.

Line114 : Graphite did not involve in isotopic exchange process? Mass imbalance?

Line 119: What is error bar for experimental data in Fig 2A?

Line 121: how do you know the pH of the run products in Fig. 2B? Do you measure?

Line 235: Ca(HCO₃)⁻ is not the only predominant phase in calcite solubility experiments.

Line 704 If most of CO₂ are from oxidation of graphite, its carbon isotopic value should be low.

I am looking forward seeing this interesting study published.

Best, Zhang

For clarity, we summarize here the revisions we made according to other major points remarked by the reviewers. Text lines cited below are referring to the main manuscript file “Tumiati_NatComm_rev1_def.pdf”. All the modified parts are highlighted in the file “Tumiati_NatComm_compare.pdf”.

Reviewer #1:

I think this is an extremely interesting study

We thank the reviewer who fully recognized the importance of our work. All comments and suggestions addressed by the reviewer have been considered and integrated in this improved version of the manuscript.

I think already starting from the abstract there should be more emphasis on the impact of your study on our understanding of volcanic gas emissions and the global carbon cycle... As written now, abstracts and the final part of the manuscript do not fully highlight the importance of your study. Your study has also important implications for a very intense discussion ongoing around the impact of crustal carbonate assimilation on the intensity of volcanic eruptions (e.g. Freda et al., 2010; Troll et al., 2012). In particular, the C isotopic signature of volcanic gas emission is used as an evidence for assimilation of upper crustal carbonates, increased CO₂ content of the magmas, which therefore can produce more explosive eruptions.

The organization of the last paragraphs has been modified, and the abstract (lines 29-51) and the paragraph “*The link between sedimentary slab fluids and arc emissions*” (now the concluding one) have been almost entirely rewritten (lines 343-380) including the proposed references Freda et al. (2010) and Troll et al. (2012). In particular, we suggest that “one major implication is that fluctuations in $\delta^{13}\text{C}_{\text{CO}_2}$ do not necessarily reflect variations in the source input, for instance due to shallow crustal carbonate assimilation or to burial of anoxic sediments dominated by organic matter. It could reflect instead local variations in fluid-rock ratios (i.e., channelized vs. percolating flux) or in the redox state (i.e., oxidizing vs reducing conditions).”. This does not exclude a priori the assimilation of crustal carbonates and its role in shifting the $\delta^{13}\text{C}_{\text{CO}_2}$ towards high values. In some areas, as those cited by the reviewers and evidenced also by Mason et al. (2017), evidence of crustal assimilation is fairly convincing. In our study, we stress that $^{13}\text{C}_{\text{CO}_2}$ higher than the global average arc isotopic composition (the dark blue surface above the orange box as depicted in Figure 4) can be

alternatively related to a set of redox state and fluid-rock ratio values, originated from the interaction of fluids coming from the dehydrating slab with the sedimentary pile of the slab where both calcium carbonate and graphite occur as reported in a variety of modern deep ocean sediments. Following our mathematical model (Fig. 4) $\delta^{13}\text{C}_{\text{CO}_2}$ higher than the average arc can be obtained for very low fluid-rock ratios and/or relatively reducing conditions close or below FMQ. Actually, at these conditions the production of $^{12}\text{-C}$ rich CO_2 by redox dissolution of organic matter (deep red in Figure 4) is hampered, and therefore the isotopic system is dominated by carbonate, as it would be in the case of crustal carbonate assimilation.

Reviewer #2:

The experiments design is novel...If this is a kinetic process, I am wondering if we can apply the experimental results to interpret natural process in geological time.

Indeed the specific design adopted in this work is entirely novel. It however relies on a combination of consolidated experimental protocols, for instance the capsule-piercing techniques which has been developed in our laboratory (Tiraboschi et al., 2016 Geofluids; Tumiati et al., 2017 Nat Comm; Tumiati et al. 2020 GCA; Miozzi and Tumiati, 2021 GPL). In our experiments, where we employed perfectly ordered crystalline graphite, we didn't observe evidence of isotopic reequilibration even in the longest 240-h experiments, but we observe CO_2 production comparable to thermodynamic predictions (Fig. 2A). Graphite is chemically reactive because the surface of the flakes is continuously in contact with the interacting fluid, and this layered oxidation process will go on until the fluid becomes saturated in carbon (Fig. 3C). The isotopically inert behavior of the inner, non-oxidized volume of the graphite flakes is remarkable but not new. It must be considered that this build on a robust background (cf. lines 63-65). For instance, Kitchen & Valley (1995) observed very slow isotopic diffusion even in medium grade natural rocks, and the maximum diffusion distance is reported to be 0.01-0.13 μm assuming 100 My period and 1000°C . At 700°C , the diffusion coefficient would be even lower, in the order of $10^{-31} - 10^{-38} \text{ cm}^2 \text{ s}^{-1}$. We thus believe that our experimental data could be very useful for the interpretation of geologic processes occurring at subarc conditions involving graphite and calcium carbonate. We discussed this concept and in general issues related to graphite dissolution in nature in the last paragraph (lines 360-364).

For the oxidation reaction of graphite (Eq. 2), We think O₂ is not existing in water-rock system at high pressure in subduction zone, so what is the oxidizing medium for this reaction?

Of course the concentration of the species O₂ is vanishingly small in the deep Earth's interior (cf. Tumiati et al., 2015 Lithos). Nonetheless, it is widely recognized that variations in redox state of metasomatized mantle and of subducted lithologies are somewhat related to variable contents of oxygen as a thermodynamic component and, consequently, variable oxygen budget (e.g., Evans et al., 2012 Geology). The driving force that controls the reaction is actually the chemical potential of O₂ (i.e. fO_2). We reformulate the sentence indicated by the reviewer emphasizing the fO_2 dependence of graphite oxidation reaction (lines 227-228). In our experimental setting, the medium is water and the graphite oxidation is controlled by the chemical potential of H₂, which is permeable across the walls of the inner capsule, fixed by the FMQ buffer in the outer capsule (lines 392-395). In nature processes are likely more complex and how fluids coming from the underlying subducted oceanic crust can impose their oxygen chemical potential to metasediments must be considered. In our conceptual model, aqueous fluids equilibrate their fO_2 with the mineral assemblages of the eclogitized oceanic crust beneath (or mixed with in the case of a subduction mélange) the sedimentary pile. Fluids therefore transfer oxygen (as a thermodynamic component) from the mafic rocks, characterized by a relatively high fO_2 and redox budget (Tumiati et al. 2015, Lithos), towards the sedimentary pile, prompting graphite oxidation. Therefore, the concluding sentence of the manuscript (lines 378-380) introduces now as a future perspective the importance of the link between chemical potential of oxygen and redox budget (i.e., the amount of oxygen as a thermodynamic component): "It remains an open question how these oxygen fugacity values correlate with the amount of oxygen transferred from the slab to the IAB source, with consequences for the redox budget of subarc mantle".

We repeated the DEW modeling in CaCO₃ and water system, and found at least the CO₂(aq), H₂CO₃(aq) are in same concentration level as Ca(HCO₃)₂

We checked again the DEW model calculations, and we confirm the results shown in Supplementary Table 3. Although it is not easy to explain why the reviewer obtained different results, we may guess that there is an issue related to the activity coefficients used for the neutral CO₂ species. In our calculations, we followed Huang and Sverjensky (2019) who recommended using relatively large activity coefficients for CO_{2,aq}. The activity coefficients for CO_{2,aq} that we used, resulted in our low concentration for the species CO_{2,aq}. If the reviewer had used an activity

coefficient of unity, that would have led to high concentrations of $\text{CO}_{2,\text{aq}}$, which would explain the difference between their result and ours. We did not include $\text{H}_2\text{CO}_{3,\text{aq}}$ in our model owing to the paucity of data characterizing this species in Huang and Sverjensky (2019). We wish to also emphasize that, in our study, the concentration of neutral CO_2 has been measured with quadrupole mass spectrometry, too, and it fits well with our calculations in both the systems aragonite + water and graphite + water (lines 114-143).

The carbon isotopic composition of volcanic arc emission around Pacific Ocean are totally heterogeneous from west to east due to input different ratio of carbonate and organic carbon (Plank and Manning, 2019).

Our experimental work challenges the traditional assumption that the carbon isotopic signature of gaseous arc emissions reflects the bulk isotopic composition of the sedimentary source. In particular, we demonstrated that the isotopic composition of arc CO_2 is non-dependent on the organic/inorganic carbon ratio of the subducted sediments (cf. *Abstract* – lines 38-41; *The link between sedimentary slab fluids and arc emissions* – lines 364-368). Instead, it is dependent on the ratio between i) the CO_2 produced by graphite oxidation and ii) calcium carbonate, which acts as an isotope buffer due to the exchange reaction $\text{Ca}^{13}\text{CO}_3 + {}^{12}\text{CO}_2 = \text{Ca}^{12}\text{CO}_3 + {}^{13}\text{CO}_2$ (lines 227-243). Therefore, for a given amount of calcium carbonate, the two variables controlling this process are only those controlling graphite oxidation to CO_2 , which are: 1) the redox state of the environment (affecting the $X_{\text{CO}_2} = \text{CO}_2/(\text{H}_2\text{O}+\text{CO}_2)$ ratio of the fluid in equilibrium with graphite) and 2) the fluid-rock ratio (affecting the absolute amount of CO_2 produced by oxidation of graphite, assuming constant X_{CO_2} because of thermodynamic considerations). Therefore, we challenged also the interpretation that local heterogeneities of arc CO_2 isotopic composition reflect the bulk organic/inorganic ratios in the source. In our experiments the measured CO_2 is actually enriched in heavy ^{13}C “despite the overwhelming abundance of graphite in the starting materials (graphite/aragonite(molar) = 5.6” (line 311-313). We suggest that local heterogeneities “could reflect instead local variations in fluid-rock ratios (i.e., channelized vs. percolating flux) or in the redox state (i.e., oxidizing vs reducing conditions)” (lines 370-372). Our conclusion sheds some light to the question raised by Plank and Manning (2019) about the reason why arc C values are skewed towards high $\delta^{13}\text{C}$ (carbonate-like) values. In our interpretation, this is not due to “preferential recycling of carbonate to the arc and preferential subduction of more refractory organic carbon to the deeper mantle (Plank and Manning, 2019)”, but to ΔFMQ values ranging from +0.78 to +1.08 that are “predicted assuming a fluid–rock mass ratio of 0.01–0.06 (=0.05–0.33

molar), which has been suggested for infiltrating, non-channelized fluids produced by dehydration reactions in eclogites. These oxygen fugacity values agree with estimations of island-arc-basalt (IAB) sources and supra-subduction mantle peridotites equilibrated at subarc depths” (lines 374-380). We stress that Figure 4 is evocative in this sense. The intersection of the fluid-rock ratio values resulting from the dehydration of oceanic crust assuming porous flow (green surface in Figure 4) and the global average arc $\delta^{13}\text{C}_{\text{CO}_2}$ (orange box in Figure 4), provides hints for the redox state in the sedimentary carbon source (yellow field; preferred model in Figure 4). However, $\delta^{13}\text{C}_{\text{CO}_2}$ higher (dark blue; controlled by carbonate) or lower (dark red; controlled by organic matter) than the average arc $\delta^{13}\text{C}_{\text{CO}_2}$ are possible for a wide set of redox state and fluid-rock ratio values, without invoking a change in the organic/inorganic carbon composition of the source.

Minor comments:

Line 75: recycled sedimentary carbonate + mantle carbon = arc magmas?

Following Mason et al. (2017) and Plank and Manning (2019), we assume that the carbon emitted by arc magmatism is derived from subducted sediments, neglecting the contribution of the mantle reservoir.

Line 89: pure graphite cannot represent organic matter, there are some hydrocarbon compound in organic matter, which may be stabilized at 3 GPa and 700 oC.

Hydrocarbon and other organic compounds can be generated starting from graphite + H₂O if P, T and $f\text{O}_2$ conditions are appropriate (e.g., Li, 2016, GPL; Tumiati et al., 2017, NatComm), so these compounds do not necessarily need to be added in the runs since the beginning. In the present study, however, we checked but didn't find evidence of organic compounds (for instance ethane) in the volatile phase.

Line 114 : Graphite did not involve in isotopic exchange process? Mass imbalance?

Please refer to the first point addressed above.

Line 119: What is error bar for experimental data in Fig 2A?

Typical analytical uncertainty is 1% (line 710).

Line 121: how do you know the pH of the run products in Fig. 2B? Do you measure?

Measuring pH in piston-cylinder experiments is currently not feasible. The pH has been calculated with the DEW model (line 713).

Line 235: Ca(HCO₃)- is not the only predominant phase in calcite solubility experiments.

It is, according to our calculations. Please refer to reply to the third point above.

Line 704 If most of CO₂ are from oxidation of graphite, its carbon isotopic value should be low.

This is not necessarily true; this is the major finding of our study. Please refer to the second point addressed above.

REVIEWER COMMENTS

Reviewer #2 (Remarks to the Author):

On the whole, we are satisfied with responses to most of our concerns in the revision. Authors found that the isotopic composition of arc CO₂ is non-dependent on the organic/inorganic carbon ratio of the subducted sediments, which does challenge the traditional assumption. We suggest accepting to publish it in the current version of the manuscript, although I am still concerned about the kinetic process for isotopic exchange during experiments.